# Advancements in Cerebrospinal Fluid Biosensors: Bridging the Gap from Early Diagnosis to the Detection of Rare Diseases

**DOI:** 10.3390/s24113294

**Published:** 2024-05-22

**Authors:** Ghazal Hatami-Fard, Salzitsa Anastasova-Ivanova

**Affiliations:** The Hamlyn Centre, Imperial College London, South Kensington Campus, London SW7 2AZ, UK

**Keywords:** CSF, biosensing, smart sensors, minimally invasive

## Abstract

Cerebrospinal fluid (CSF) is a body fluid that can be used for the diagnosis of various diseases. However, CSF collection requires an invasive and painful procedure called a lumbar puncture (LP). This procedure is applied to any patient with a known risk of central nervous system (CNS) damage or neurodegenerative disease, regardless of their age range. Hence, this can be a very painful procedure, especially in infants and elderly patients. On the other hand, the detection of disease biomarkers in CSF makes diagnoses as accurate as possible. This review aims to explore novel electrochemical biosensing platforms that have impacted biomedical science. Biosensors have emerged as techniques to accelerate the detection of known biomarkers in body fluids such as CSF. Biosensors can be designed and modified in various ways and shapes according to their ultimate applications to detect and quantify biomarkers of interest. This process can also significantly influence the detection and diagnosis of CSF. Hence, it is important to understand the role of this technology in the rapidly progressing field of biomedical science.

## 1. Introduction

Examination of the cerebrospinal fluid (CSF) is an important diagnostic tool for numerous cancers, immunological disorders, and neurodegenerative diseases [1,2,3]. Analysis of CSF dates back to 1764 [4], and further detection of neurodegenerative diseases from CSF gained attention in the 1990s [5,6]. Since then, investigations of CSF analysis for the early diagnosis of various neurological conditions and malignancies have gained significant importance. The investigation of the components and quality of CSF is similar to blood analysis tests for the neurological system [1]. Since the nervous system lacks lymphatic drainage, the CSF acts as the lymphatic system of the brain and nervous system [7]. Hence, it is responsible for the transport of neurotransmitters and toxins from the central nervous system (CNS). However, CSF analysis is limited to the detection of specific individual biomarkers [8]. Since CSF conditions are relatively stable, any deviations in its content from the normal range can be a warning for neurological disease. The low concentrations (trace level) of a few biomarkers limit the detection of analytes by analytical laboratory tools [5] in the early stages of diseases. Laboratory analytical tools, such as mass spectroscopy [9], ELISA [10], Western blotting [11], and HPLC [12], are well-known techniques for analyte detection. However, these techniques require time for sample preparation and pretreatment, in addition to being expensive. Hence, the most important challenges are fast detection, high accuracy, and increased sensitivity for in situ analyte detection in CSF. The enzyme-linked immunosorbent assay (ELISA) is a widely applied method in clinics and hospitals across the world for accurate detection and quantification of biological agents, mainly proteins and polypeptides [13]. It offers a wide range of detection; is linear, quantifiable, and easy to use; and provides multiple output signals with different fluorophores. However, it may not be as sensitive as chemiluminescence and film, and the cost of an imager can be a disadvantage [13]. Western blotting is a technique known for its sensitivity. It can detect as little as 0.1 nanograms of protein in a sample, making it an effective early diagnostic tool [14]. It offers a wide range of detection; is linear, quantifiable, and easy to use; and provides multiple output signals with different fluorophores. However, it may not be as sensitive as chemiluminescence and film, and the cost of an imager can be a disadvantage. High-Performance Liquid Chromatography (HPLC) is a technique used for separating, identifying, and quantifying each component in a mixture of compounds [15]. It is widely used in biochemical fields for the detection of small molecules like metabolites, drugs, and biomarkers in body fluids. HPLC offers high resolutions, sensitivity, and precision, and it allows for the analysis of a wide range of samples. However, despite its advantages, HPLC can be costly, requiring large quantities of expensive organics. Techniques such as solid-phase extraction and capillary electrophoresis can be cheaper and even quicker, especially for analysis under good manufacturing practice. Poor resolutions and inconsistent recoveries are notable drawbacks of this method when crude unprocessed samples are used.

However, the specifics of its advantages and disadvantages in detecting biomarkers in CSF are not well documented and may require further research.

This review article aims to investigate the recent improvements in non-invasive, point-of-care, CSF-sensing, and monitoring systems and highlights challenges and future perspectives.

### 1.1. CSF’s Formation, Mechanism, and Function in the Body

Ependymal cells present in the choroid plexus (CP), which line the floors of the lateral ventricles and the roofs of the third and fourth ventricles of the brain, are predominantly responsible for CSF production [16]. Volumes of 0.2–0.7 mL/minute of CSF are generated, mostly within the lateral ventricles of the brain. About 400–600 mL of CSF is generated approximately four times a day [17]. CSF is a colourless, acellular, low-protein fluid [1]. It contains approximately 60–80% glucose, 22–38 mg/dL protein, and less than five mononuclear cells per mm^3^ in healthy adults [18]. CSF is a part of the extracellular fluid that contains various small molecules and proteins. CSF also plays a crucial role in cerebral blood flow regulation [19]. Carrying essential ions, CSF maintains the healthy functioning of the central nervous system. Cerebrospinal fluid is a reason for the buoyancy of the brain and reduces its weight to 25–50 g from its actual 1000–1500 g weight [20]. It also acts as a barrier to protect the brain and spinal cord from sudden shocks and damage [21]. Furthermore, toxins produced in the brain are transported to the bloodstream via CSF. The exchange of small and large molecules between the blood and CSF is controlled by the blood–CSF barrier [22]. It is also involved in nutrient exchange between the blood and nerves and provides a crucial environment for neuronal signalling. Hence, any long-term disruption of this regulation may result in a malicious disease [23]. CSF is a crucial part of the Monro–Kellie doctrine, where raised intracranial pressure is detected by an increase in CSF pressure [24]. Figure 1 shows CSF collection and analysis procedures.

### 1.2. CSF Collection and Challenges

To diagnose any disorder in the central nervous system through CSF screening, a patient sample should be collected and analysed. CSF sampling has been performed via needle biopsy or lumbar puncture (LP) for many years [25]. LP is an invasive procedure that enables CSF sampling in the spinal subarachnoid space. Depending on the sampling site, the components of CSF, such as plasma proteins, may have varied analyte concentrations [26].

In the early 1890s, LP was introduced as an intracranial pressure release treatment for meningitis [27]. However, this procedure has been predominantly used as a diagnostic tool [28,29]. Lumbar puncture has been used for CSF sampling and anaesthesia for years, but it has certain risks and leads to post-procedural complications. Complications caused by or through LP can occur immediately or after the procedure. One of the most common symptoms was reported to be a long-lasting severe headache after LP. This reported morbidity could result in fatality if not treated [30]. As lumbar puncture is performed via needles of different sizes, according to research investigations, a smaller needle can significantly decrease the risk of post-lumbar puncture headaches [31]. However, as the needle penetrates the subarachnoid space, it can act as a vehicle for microorganisms to cause local infections. This invasive sampling and screening method can result in inflammation or infection of the patient’s body [32]. The risk of infection increases if the patient has skin lesions or dermatological conditions, such as psoriasis. This is mostly due to the risk of S. aureus infection in patients with psoriasis and other dermatological conditions [33]. Increased risk of haemorrhage leading to a spinal haematoma and the potential for brain herniation due to raised intracranial pressure are also known to be possible complications of this procedure [32]. The overall risk of complications after lumbar puncture can be more severe in infants [34]. Despite these risks and complications during LP, it remains the primary method for anaesthesia and CSF sampling [25,35].

Given that LP remains the chief method used to study CNS disorders, diagnostic tools have been developed to better analyse CSF [36]. Furthermore, the rapid development of novel non-invasive biomedical devices has allowed the properties of CSF to be widely studied in recent decades [37].

### 1.3. Changes in CSF Associated with Various Diseases

As mentioned above, the footprint of any CNS (central nervous system)-associated or infectious disease can be detected in the CSF. As the CSF is in contact with neuronal cells, choroid plexuses, and the blood–brain barrier membrane, it partially contains molecules or peptides secreted from them [17,38]. Exposure of the CSF to various fungal, bacterial, viral, and parasitic infections can also be detected. Hence, alterations in the CSF can be used as tools for early diagnosis, treatment, and intracranial pressure monitoring. The CSF components in a healthy adult under normal conditions are shown in the table below (Table 1).

Accordingly, any change in the physiological state of the body can cause these values to deviate from their normal ranges.

Infections affecting CSF quality may originate from various sources. Viruses infect the CNS to transmit their genetic material (DNA or RNA) to host cells for self-replication [49]. Viral encephalitis and meningitis can cause severe inflammatory responses that can be detected in the CSF. Inflammatory biomarkers are detectable a few hours to days after infection [50]. Decreases in CSF glucose levels have been reported [51]. Detection of viruses is possible through a polymerase chain reaction (PCR) of CSF samples [52].

Bacteria can infect the central nervous system through bacteraemia, a wound in the body, or the course of a lumbar puncture [53,54]. One of the worst types of CNS infections is bacterial meningitis, which can be fatal [55]. After infection, substantial numbers of pro-inflammatory markers appear in the CSF, and the type of infection can be identified using PCR [56]. In the case of bacterial infections, in some cases, detection of bacteria in CSF samples is possible via microscopy.

Fungal infections, including moulds, yeasts, and certain fungal strains, can also penetrate the blood–brain barrier (BBB) and infect the CNS, leading to severe conditions such as meningoencephalitis, meningitis, and cerebral abscess [57]. The white blood cell count usually increases in the CSF after fungal contamination, but the type of white blood cell (Monocytes, Eosinophils, etc.) is mostly dependent on the strain of fungal contamination. Reduced glucose levels and increased protein levels have also been reported in fungal infections of the CNS; however, the distinction of fungal CNS contamination from other categories has been a challenge due to non-specific signs [58]. In many low-income to middle-income countries, parasitic infections (e.g., Malaria, Leishmaniasis, Amebiasis, Hydatidosis, Coenurosis, and Sparganosis) are among the causes of CNS diseases [54]. Unlike fungal infections, parasitic infections of the CNS have a few symptoms, such as seizures, eosinophilia, headaches, and fever [59]. Parasitic infections in CSF samples can be detected using microscopy and PCR [54].

Any of the abovementioned means of infection can lead to severe conditions involving the central nervous system. Conditions that arise from infection sources can be acute or chronic. Severe acute or chronic infections of the CNS can be starting points for neurodegenerative conditions. Acute conditions of the CNS can usually alter the quality and content of the CSF [60].

Neurodegenerative disorders can affect the CSF content. The onset of each infection or disorder affects the CSF content differently. Initially, inflammatory markers appear in the CSF in mild cases of inflammatory infections [61]. The total protein count in the CSF increases depending on the severity of the inflammation [62]. Disorders such as Parkinson’s disease, myelopathies, and neuropathies are included in this category [63,64,65].

Peripheral neuropathy is a common neurological disorder caused by various factors such as diabetes or hypothyroidism [66]. Neuropathies are diagnosed based on various symptoms and analyses, including CSF analysis. They are characterised by increased inflammatory markers and protein counts along with the physical symptoms of the patient. Furthermore, an increase in small molecules and metabolic materials in CSF white blood cells can be detected in diseases such as globoid cell leukodystrophy and mucopolysaccharidosis [67].

Subarachnoid haemorrhage is a condition that has a direct effect on CSF, usually a few hours after an incident or trauma [68,69]. This condition is painful and can sometimes be followed by a loss of consciousness. In this case, the red blood cells found in the CSF can be detected at >1000 CFU/mm [70]. CSF screening is a well-known approach used to investigate meningitis. Meningitis is inflammation of the protective membrane of the brain and spinal cord. As mentioned above, the root cause of meningitis can be a viral or bacterial infection. It can be initiated by bacterial, fungal, viral, or protozoan microorganisms. CSF analysis is a crucial tool, particularly in the diagnosis and treatment of aseptic meningitis [71]. However, it may take days to obtain the results of CSF analysis using PCR. Analysis of the CSF in aseptic meningitis can identify the source of the infection [71]. For instance, a bacterial infection increases the turbidity of the CSF, whereas viral and fungal infections do not have a significant effect on turbidity. On the other hand, in aseptic meningitis, the protein content of the CSF increases. Viral and fungal infections usually have fewer indications for diagnosis in the CSF; however, increases in white blood cells and lymphocytes in the CSF can be warning signs. Overall, a PCR test can be conducted on a CSF sample to indicate the infection type and the organism that has caused the infection.

Detection of autoimmune disease markers in CSF can offer early treatment opportunities. In general, the early signs of autoimmune diseases such as multiple sclerosis commence with the detection of increased protein levels and lymphocytes in the CSF. Elevated protein levels are a common sign of most autoimmune diseases. In some cases, such as transverse myelitis, inflammatory markers are detectable in CSF. Autoimmune Encephalitis (AEI) is an inflammatory disease associated with the central nervous system. The findings from a CSF analysis of patients with AEI suggest that the inflammation and antibody markers of AEI do not follow certain patterns in various patients. However, anti-NMDA, GABAb, and AMPA antibodies have shown inflammatory changes in CSF samples from patients [72].

Brain and spinal cord cancers also affect CSF quality, which can be a promising tool for diagnosing and preventing metastasis [73]. Reports suggest that CSF analysis, along with MRI and CT scans, is a promising tool for detecting malignant tumours. This method can be expanded to the detection of colon, lung, and prostate cancer biomarkers [74].

Furthermore, minute changes in spinal cord injuries have sensible effects on CSF quality and biomarker contents. As a result of spinal cord tissue damage, neurofilaments (NFs), tau, neuron-specific enolases (NSEs), S100 calcium-binding protein β (S100β), and glial fibrillary acidic protein (GFAP) can eventually be released into the CSF [75]. Along with the vast number of studies [76] performed on Glial Scarring in the spinal cord, the detection of CSPGs, MAG, Nogo-A, and Omgp in CSF can also be investigated in the category of spinal cord defects.

Overall, changes in CSF biomarkers can facilitate the diagnosis and prognosis of various neurological, inflammatory, and infectious conditions. The table below (Table 2) illustrates the most important biomarkers in CSF that can be associated with the most common neurological diseases.

## 2. An Overview of Biosensors

Biosensors are analytical devices that combine a biological component with a physicochemical detector. The term “biosensor” is often used to cover sensor devices used in order to determine the concentrations of substances and other parameters of biological interest, even where they do not utilise a biological system directly [89]. These devices can provide fast, reliable, and accurate detection of specific biological processes and analytes. The biological components, also known as bioreceptors, can be tissues, microorganisms, organelles, cell receptors, enzymes, antibodies, nucleic acids, etc. The bioreceptor interacts with, binds with, or recognises the analyte being studied (Figure 2). The transducer or detector element works in a physicochemical manner, transforming one signal into another [90].

### 2.1. Electrochemical Biosensors

Electrochemical biosensors are self-contained integrated devices that provide specific quantitative or semi-quantitative analytical information using a biological recognition element (biochemical receptor) held in direct contact with an electrochemical transduction element [91]. They use biological materials as sensitive components; electrodes as conversion elements; and current, potential, and resistance as the detectable characteristics of a signal. In recent decades, electrochemical biosensors have been developed to detect many biological elements. They offer advantages such as high sensitivity, robustness, reliability, and the potential for integration on a single chip [92].

### 2.2. Optical Biosensors

Optical biosensors use light and biological elements to detect the presence of an analyte. They primarily use enzymes and antibodies as transducing elements. Optical biosensors allow the safe non-electrical remote sensing of materials. They usually do not require reference sensors, as a comparative signal can be generated using the same light source as the sampling sensor [93]. They have exhibited good performance in detecting biological systems and have promoted significant advances in clinical diagnostics, drug discovery, food process control, and environmental monitoring [94].

### 2.3. Piezoelectric Biosensors

Piezoelectric biosensors use the piezoelectric effect to measure changes in pressure, acceleration, temperature, strain, or force by converting them to electrical charges. They provide a direct method for the real-time monitoring of biointeractions at the sensor surface, resulting in simplified assay formats [95]. They are well suited for the construction of biosensors that recognise affinity interactions. Assays based on reactions between antigens and antibodies, two polynucleotide strains, aptamers, and proteins are well suited for such studies [96].

### 2.4. Amperometric Biosensors

Amperometric biosensors function by producing a current when a potential is applied between two electrodes. They generally have response times, dynamic ranges, and sensitivities similar to potentiometric biosensors. The simplest amperometric biosensor in common use involves the Clark oxygen electrode. They provide a direct method for the real-time monitoring of biointeractions at the sensor surface, resulting in simplified assay formats [97].

### 2.5. Voltametric Biosensors

Voltametric biosensors are electrochemical biosensors that measure the current (flow of electrons) arising during a reaction. Generally speaking, when using voltametric biosensors, the analyte undergoes or is involved in a redox reaction that can be followed by measuring the current in an electrochemical cell [98]. They are often used in initial analytical tests because of their characteristics such as ease of understanding (simple theory), ease of use (choice of potential and current acquisition time), and the possibility to work with very simple and low-cost equipment [98].

### 2.6. Development of Biosensors

Biosensor development involves several steps. The first step involves the selection of a suitable biological sensing element that can interact with the analyte of interest to generate a signal [99]. Sensing elements include materials such as tissues, microorganisms, organelles, cell receptors, enzymes, antibodies, and nucleic acids. The signal generated through the interaction of the sensing element and the analyte of interest is then transformed into a measurable and quantifiable electrical signal via the transducer [100].

Modern approaches employed in biosensors involve the use of nanomaterials, such as nanoparticles (NPs), nanowires (NWs), nanorods (NRs), carbon nanotubes (CNTs), quantum dots (QDs), and dendrimers. These nanomaterials can enhance the performance of transducers, increase sensitivity, reduce response time, improve reproducibility, and lower detection limits [90].

### 2.7. Applications of Biosensors

Biosensors are used in a wide range of applications. They play critical roles in a myriad of fields, including biomedical diagnosis, monitoring of treatment and disease progression, drug discovery, food control, and environmental monitoring [89]. They have proven useful in disease diagnosis in human healthcare delivery, agriculture, homeland security, food security, environmental and industrial monitoring, and bioprocessing [101]. They are also used in general healthcare monitoring, screening for diseases, the clinical analysis and diagnosis of diseases, veterinary and agricultural applications, industrial processing and monitoring, and environmental pollution control [102]. Biosensors represent a promising area of research with the potential to revolutionise the diagnosis and monitoring of various diseases. As this technology matures, it is expected to become an integral part of community-based screening programs and personalised medicine.

## 3. Types of Biosensors

Three generations of biosensors are available on the market. Furthermore, five types of sensitive elements are used: antibodies, nucleotides, enzymes, cells, and synthetic molecules [103]. Numerous biosensors have been constructed in many forms, including electrochemical and fluorescence-tagged nanomaterials, silica or quartz, and genetic constructs with reporter genes [104].

Depending on the type of transducer used in biosensor design, biosensors can be categorised into optical biosensors, electrochemical biosensors, or mass-based biosensors.

Comparing different types of biosensors is crucial for several reasons. Each type of biosensor, whether it is an immunosensor, enzymatic sensor, peptide-based sensor, or gene-based sensor, has unique advantages and applications [105]. By comparing these biosensors, we can understand their strengths and limitations, which can guide the selection of the most appropriate biosensor for a specific application. Furthermore, such comparisons can inspire the development of new biosensors that combine the advantages of different types, leading to improved performance and expanded applications. Therefore, comparing different biosensors is an essential step in the ongoing advancement of biosensor technology.

### 3.1. Immunosensors

Immunosensors are biosensors that utilise antibodies as biological recognition elements. They are popular because of their excellent detection performance, selectivity, and sensitivity. Recent advancements in immunosensors allow detection to be incorporated in the latest digital technology and miniaturised without compromising performance. They are quick, highly selective, and sensitive and possess the potential to significantly improve the diagnostic processes of pathogens and toxins [106]. They are a promising alternative to the traditional immunoassays and state-of-the-art affinity sensors used to diagnose clinically important analytes/antigens due to their high affinity, versatility, compact size, fast response time, minimum sample processing, and reproducibility [107].

### 3.2. Enzymatic Biosensors

Enzymatic biosensors use enzymes as biological recognition elements. They are highly specific, simple, and scalable and facilitate the fast, precise, and continuous monitoring of analytes. The high specificity of enzymes enhances their ability to detect analytes with lower concentration limits [108]. They provide a direct method for the real-time monitoring of biointeractions at the sensor surface, resulting in simplified assay formats. They generally have response times, dynamic ranges, and sensitivities similar to those of potentiometric biosensors [109].

### 3.3. Peptide-Based Biosensors

Peptide-based biosensors use peptides as biological recognition elements. They have been rapidly developed for the detection of disease markers with high sensitivity, rapid analysis speeds, and easy miniaturisation [110]. Their benefits mainly lie in their stability and selectivity toward a target analyte. Furthermore, they can be synthesised easily and modified with specific functional groups, thus making them suitable for the development of novel architectures for biosensing platforms as well as alternative labelling tools [111].

### 3.4. Gene-Based Biosensors

Gene-based biosensors, also known as DNA-based biosensors, use DNA or RNA as a biological recognition element [112]. They have advantages such as wider detection targets, longer lifetimes, and lower production costs. Additionally, ingenious DNA structures can control the signal conduction near the biosensor surface, which could significantly improve the performance of the biosensors. They can be used effectively to provide simple, fast, cost-effective, sensitive, and specific detection of certain genetic diseases, cancers, and infectious diseases [113].

## 4. Multiplex Biosensors

Multiplex biosensors are an emerging approach to point-of-care testing. They reduce analysis time and testing costs by detecting multiple analytes or biomarkers simultaneously. These are crucial for disease detection at an early stage. The application of inexpensive substrates such as paper has immense potential and is a matter of research interest in point-of-care testing for multiplexed analysis [114]. These biosensors possess several unique advantages, including the ability to detect multiple analytes on one instrument, examine various parameters, and offer a 2D paper network with spatiotemporal complexity [115].

### 4.1. Advancements in Multiplex Biosensors

Recent advancements in biotechnology and nanotechnology have facilitated the development of biosensor platforms capable of real-time detection of multiple biomarkers in clinically relevant samples [116]. Both electrical and spectroscopic multiplexed detection strategies are employed to reach limits of detection below the levels in clinical samples [116]. Some of the most promising strategies for point-of-care assays involve inexpensive materials, such as paper-based microfluidic devices, or portable and accessible detectors, such as smartphones [117].

### 4.2. Multiplex Biosensors in Cerebrospinal Fluid (CSF) Biosensing

In recent years, biomarkers of inflammatory responses in the central nervous system (CNS) have been identified in different body fluids, such as blood, cerebrospinal fluid (CSF), and tears [118]. Progress in micro- and nanotechnology has enabled the development of biosensing platforms capable of detecting multiple biomarkers in clinically relevant samples, such as CSF. These advancements have opened new possibilities for the diagnosis and monitoring of neurological diseases [118].

For instance, an electrochemical biosensor has been reported for the highly sensitive label-free detection of CSF miR-21, relying on target-induced redox signal amplification [119]. This biosensor was developed by covalently assembling capture stands partially complementary to miR-21 on a gold nanoparticle-coated glassy carbon electrode [120].

## 5. Biosensing in Body Fluids

Detection of central nervous system (CNS) disease biomarkers using biosensors in various body fluids has emerged as a promising approach in the field of neurology. These biosensors have been developed to detect key biomarkers, such as brain-derived neurotrophic factor (BDNF) and neurofilament light chain (NfL), which play important roles in the development and progression of many neurological diseases, including multiple sclerosis, Alzheimer’s disease, and Parkinson’s disease [121].

### 5.1. Saliva-Based Biosensors

Saliva-based biosensors are gaining attention owing to their non-invasive collection and ability to detect periodontal disease and identify systemic diseases. The human body has a unique way of indicating when something is wrong [122]. Molecules in body fluids can be helpful in the early detection of diseases by enabling health and preventing disease progression. For instance, C-reactive protein (CRP) is the most predictive biomarker of Acute Myocardial Infarction [123]. Several sensing technologies are available for stress biomarker monitoring in saliva and other fluids. Enzyme-linked immunosorbent assays, colourimetric techniques, surface plasmon resonance sensing, and molecularly imprinted polymers offer sensitive and selective cortisol detection in saliva.

### 5.2. Blood-Based Biosensors

Blood-based biosensors have gained attention because of their ability to detect a wide range of biomarkers for various diseases. Alzheimer’s disease (AD) is a chronic neurodegenerative disorder, and its early diagnosis plays a crucial role in its prevention and treatment [124,125]. In recent years, blood-based biomarkers for early AD diagnosis have gained attention because of their non-invasiveness and minimal adverse reactions, making them an attractive approach. Among the various sensors used to detect blood biomarkers for AD, printed sensors demonstrate great potential for AD blood biomarker detection because of their ease of miniaturisation, scalability for mass production, and compatibility with various detection methods.

### 5.3. Tear-Based Biosensors

Tear-based biosensors are promising tools for detecting various biomarkers associated with neurological diseases. The TearAD study aimed to use tear fluid as a potential source of AD biomarkers. In previous reports, it was demonstrated that the AD biomarkers amyloid-beta and tau are measurable in tear fluid and are associated with disease severity and neurodegeneration [126]. This study aimed to validate the previous results in a larger cohort and evaluated the diagnostic accuracy of tear biomarkers when discriminating between individuals with and without neurodegeneration, as determined by hippocampal atrophy [126].

## 6. CSF Sensing and Detection: Current State and Progress

Recently, alternative and prompt approaches to diagnosing diseases using CSF analysis have been introduced. So far, different analytical procedures have been used in this regard, such as MRI [127], CT [128], PCR [129], GC–mass spectrometry [130], and microscopy [40,131]. As mentioned, lumbar puncture is the chief procedure for CSF sampling, which is rather painful and carries a high infection risk for patients [132]. Hence, there is a need for quick and minimally invasive techniques for CSF sampling and subsequent analysis.

Electrochemical biosensors have emerged as a practical option for minimally invasive CSF sensing [133]. Generally, a biosensor or electrochemical sensor should be both highly sensitive and selective [10] for use as a diagnostic tool. Furthermore, biosensors can be designed and characterised according to the desired application.

Since most of the problems associated with the central nervous system are delicate and difficult to address without surgery, it is important to have biosensors with suitable designs to fulfil the technical and medical requirements. For example, CSF accumulates in the brain in hydrocephalus. The conventional treatment for hydrocephalus involves implanting a shunt in the brain ventricle via surgery [134]. However, this procedure carries a high risk of failure and infection [135]. Hence, implantable sensors have been introduced as novel monitoring tools for hydrocephalus treatment and control [136]. They have been used successfully in the detection of shunt malfunctions [137,138] and CSF flow control [139]. Implantable sensors are another option for monitoring CSF pressure and have been proven to decrease the chance of infection [140]. Another continuous flow monitoring method for the detection of hydrocephalus shunt malfunctions was proposed by Pennell et al. (2016). An implantable sensor was designed to monitor and regulate shunt malfunctions. The implantable sensor was designed to uniquely characterise the type of abnormality, either blockage or decreased flow, and regulate the flow accordingly [141]. In addition to flow control, this method is less invasive and reduces the risk of infection.

The detection of biological biomarkers in CSF is an important application of electrochemical biosensors. Over the past decade, the detection of crucial biomarkers in CSF has been gaining more attention. Numerous novel approaches for detecting biomarkers in CSF samples have been introduced. For instance, biosensors integrated with microfluidics were proposed by Senel et al. in 2020 [142] to detect decreased dopamine levels in the CSF and blood. Decreased dopamine levels are a hallmark of Parkinson’s disease. Such sensors enable the detection of important biomarkers in bodily fluids. The use of a microfluidic device was also proposed as a practical approach to decrease the sample volume [142].

CSF is in contact with the cells surrounding the brain ventricles and can be exposed to variations in extracellular pH from cells, such as neuronal cells and brain tissue. The pH of CSF in a healthy adult is approximately 7.4; hence, variations in the pH of CSF can lead to an abnormal biological condition [143]. Electrochemical pH biosensors based on ZnO microneedles have successfully enabled the real-time monitoring of mouse CSF [144] in vivo. A unique fibre-based pH sensor with increased sensitivity was designed by Booth et al. in 2021 [145].

A multi-sensing system for CSF was reported to be able to detect multiple components in CSF to predict a diagnosis of Alzheimer’s disease. The sensor was successfully calibrated for dopamine, KCl, NaCl, MgCl_2_, NaHCO_3_, NaHS, and CaCl_2_ detection [146].

A non-invasive method to detect lactic acid in CSF was introduced by Goh et al. (2011). This method uses low-power microwave sensors [147]. Lactate is one of the biomarkers in CSF that is very helpful in detecting infections in the central nervous system [148]. A variety of biosensors have been investigated for the measurement of lactate in blood, plasma, interstitial fluid [149], and sweat [150] on various substrates [151]. However, a more innovative approach for point-of-care lactate sensing in CSF is yet to be developed.

So far, the main application of CSF sensing remains the early-onset diagnosis of central nervous system diseases [40]. Currently, Alzheimer’s disease and dementia are the most controversial diseases in terms of early diagnosis and treatment. Hence, early detection of the disease can be crucial for treatment progress. Proteins such as β-amyloid and tau are the most common indicators of disease onset in CSF. However, the detection of these proteins in CSF in the early stages requires highly sensitive detection techniques. Very recently, with the development of a highly sensitive β-amyloid sensor, investigators have been able to detect β-amyloid [152]. The ability of the sensor to detect β-amyloid levels as low as 1 pg/mL can increase the chance of β-amyloid detection and early-onset diagnosis. This biochemically modified sensor has the potential to be used as a detection sensor rather than the previously prepared sensors for β-amyloid detection [153]. However, β-amyloid is not the only biomarker that must be detected for an accurate diagnosis of central nervous system disease [154]. Tau protein also participates in the formation of amyloid plaques in the brain. Hence, detection of total tau protein in CSF can also have an impact on early diagnosis [155]. Analytical methods have been reported to detect tau protein [156], including ELISA, xMAP, and antibody-based assays, for the analysis of total tau protein levels [157]. However, these assays require time and analytical laboratory conditions that remain controversial. Biosensors, on the other hand, have been introduced as a fast and highly sensitive approach for tau protein detection in CSF. These approaches include optical [158], electrochemical [159], and piezoelectric biosensors [160]. Electrochemical biosensors are less sensitive than optical biosensors and more sensitive than piezoelectric sensors. Among biosensors, electrochemical biosensors have been widely utilised in various aspects of life, from healthcare to technological applications. Their flexibility of design, reliability, high limit of detection, and vast range of applications for various purposes make electrochemical biosensors a unique and wise choice [161]. This feature has also had an impact on the development of diagnostic electrochemical biosensors for applications in neurodegenerative disorders. Furthermore, electrochemical biosensors are easy to fabricate and can be designed for amperometry and voltammetry.

A list of recent advances in electrochemical CSF biosensors for neurodegenerative disease-related biomarkers and their ranges of detection are listed in the table below.

The table presents a list of recent developments in the field of CSF sensing (Table 3). Currently, valuable research is underway on the development of fast and minimally invasive in vitro diagnostic biosensors. These biosensors are aimed at detecting certain biomarkers associated with disorders in order to provide correct diagnoses.

According to the literature review, various methods have been developed to measure proteins in CSF. However, most of these methods are based on point-of-care detection via antibody–antigen bonding. Hence, there is a need for a method that is capable of continuously detecting analytes to improve repeatability. In addition to their development, the validation of these sensors in clinical settings is also important [146].

## 7. Conclusions and Future Perspectives

The detection of biomarkers in body fluids has progressed significantly, with advancements in biosensor technology playing a pivotal role. It is evident that indications of many diseases, particularly neurodegenerative diseases, can be observed in various body fluids such as cerebrospinal fluid (CSF), saliva, blood, and tears. Accurate detection of key biomarkers associated with these disorders is crucial for early diagnosis and effective treatment.

While the main and prominent collection method for CSF remains lumbar puncture, it is important to explore options to decrease its associated complications and ease the procedure. In addition, the development of fast, low-cost diagnostic devices, such as saliva-based, blood-based, and tear-based biosensors, is highly desirable for disease detection and monitoring.

Incorporating biosensors into lumbar puncture needles or designing newer needles can offer better alternative methods to achieve these goals. However, the design of these biosensors must be approached in an interdisciplinary fashion, with engineers, neurosurgeons, and biotechnologists working together to develop robust sensors with high diagnostic sensitivity and specificity.

Furthermore, the use of multiplex biosensors capable of detecting multiple biomarkers simultaneously represents a promising area of research with the potential to revolutionise the diagnosis and monitoring of neurological diseases. As this technology matures, it is expected to become an integral part of community-based screening programs and personalised medicine.

In conclusion, the future of detecting CNS disease biomarkers in various body fluids lies in the continuous advancement of biosensor technology, interdisciplinary collaboration, and the integration of these technologies into existing medical procedures and practices.

## Figures and Tables

**Figure 1 sensors-24-03294-f001:**
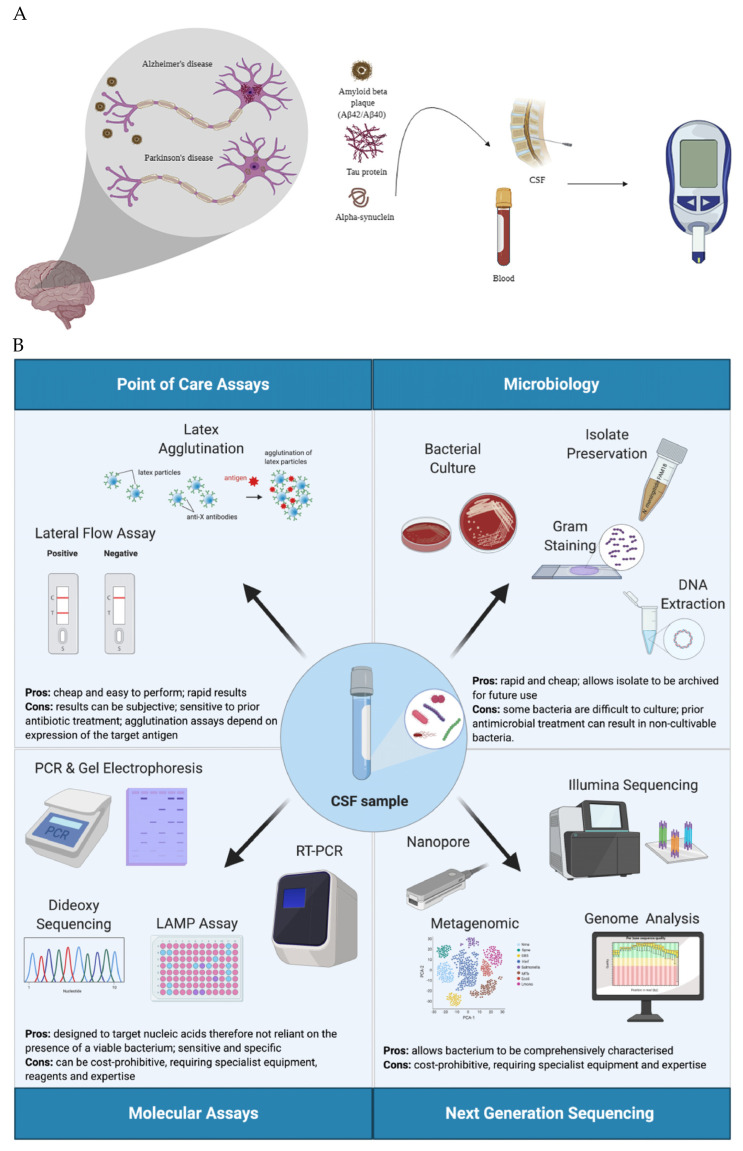
(**A**) Lumbar puncture is the most commonly used technique for CSF collection. As a biomarker of various central nervous system disorders, it is detectable in CSF [22]. (**B**) A variety of tests and techniques are available for the analysis of CSF samples, most of which are lab-based and require long processing times and special equipment. Point-of-care biosensors are the fastest approach and can be used in situ for the detection of biomarkers of interest [23]. (**C**) Electrochemical detection of analytes of interest can be carried out using a working electrode that can be modified for the detection of a particular analyte, after which a signal is transduced to a potentiostat for quantification of the concentration [24].

**Figure 2 sensors-24-03294-f002:**
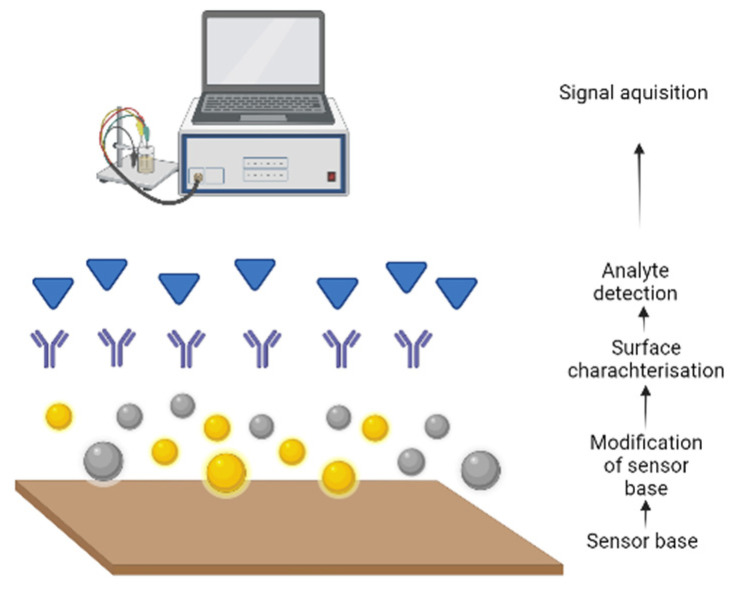
Electrochemical detection of analytes of interest (this figure was generated using biorender.com, accessed on 22 March 2024).

**Table 1 sensors-24-03294-t001:** CSF components in a healthy adult.

	Quantity	References
Colour	No colour	[39,40]
pH	7.4	[41]
Pressure	50–200 mm H_2_O	[42]
Overall cell count	<5 cells per mL	[43]
Red blood cells	0	[40]
White blood cells	0–5 (lymphocytes)	[44]
Protein	15–40 mg/dL	[45]
Glucose	50–80 mg/dL	[46]
Lactate	1–3 mmol/L	[46]
Microorganisms	No	[47]
Mg, K, Na, Ca	Trace	[48]

**Table 2 sensors-24-03294-t002:** Biomarkers associated with the most important neurological diseases.

Clinical Condition	Biomarkers in CSF	References
Alzheimer’s	Aβ1-40, Aβ1-42, tTAU, pTAU, cortisol	[77,78]
Multiple sclerosis	OCBs (oligoclonal bands) Increased IgA, IgM CRTAC, Tetranectin, autotaxin-T Immunoglobulins: Ig ϒ1, Ig heavy chain V-III, and Ig-k-chain	[79,80]
Spinal cord injuries	Pro-inflammatory cytokines in CSF, NSEs, S100β, and NFH	[81]
Guillain–Barre Syndrome	Increased CSF protein level	[82]
Amyotrophic lateral sclerosis	Total protein concentration, IL-1β, and TNF-α	[83,84]
Meningitis	Increased lactate level, CSF glucose/blood glucose < 0.4	[85,86]
Dementia	Tau protein, Aβ1-42, NF light	[87,88]

**Table 3 sensors-24-03294-t003:** Details of various biosensing platforms for various analytes in CSF.

Biomarker	Electrochemical Sensor Substrate	Related Disease	LOD	References
Tau protein	Biosensor cell lines, Sandwich-based antibody	Alzheimer’s	316 pg–100 ng 34 ng/L	[118,123]
Aβ_1-42_, Aβ_1-40,_ Amyloid-β	Dielectrophoretic force-driven MXene/multi-wall carbon printed nanotube (molecularly imprinted) Molecularly imprinted polymers and aptamers Gold-based plasmon resonance biosensors	Alzheimer’s	Pg/mL 1.0 fg mL^−1^–100.0 fg mL^−1^ 1.22 pg mL^−1^ 2.4 pg/mL	[162,163,164,165]
Dopamine	Aptamer-based biosensor Microfluidic Au-based biosensor Silica-functionalised fluorescent carbon dots	Alzheimer’s Parkinson’s Parkinson’s	µM 0.1 nM 41.2 nM	[142,164,166]
Glutamate	In vivo sensing biosensor (Pt wire-based) Carbon–Pt microparticle-based biosensor	Brain Glutamate monitoring Brain Glutamate	0.044 µM 0.03 µM	[167]
Serotonin	Reusable aptasensor (Au-based) Flexible WS2/Graphene/Polyimide Electrode-based biosensor	Alzheimer’s and Parkinson’s NA		[130]
Acetylcholine	EDOT-based solid-state biosensor Amperometry biosensors	Alzheimer’s and Parkinson’s		[131]
Glycated albumin	His6-RAGE VC1-modified electrodes used as biosensors Aptamer-conjugated magnetic nanoparticles used to precipitate glycated albumin	Alzheimer’s		[132]
Cortisol	NA	Delirium, bacterial meningitis, Alzheimer’s disease	NA	NA

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
