# Peer review of "Advancements in Cerebrospinal Fluid Biosensors: Bridging the Gap from Early Diagnosis to the Detection of Rare Diseases"

_sensors, 2024, doi:10.3390/s24113294_

Round 1
Reviewer 1 Report
Comments and Suggestions for Authors
Dear Dr. Hatami-Fard!
I have an honor to review your and Salzitsa Anastasova-Ivanova work “Advancements in Cerebrospinal Fluid Biosensors: Bridging the Gap from Early Diagnosis to Detection of Rare Diseases.”
Your work highlights the actual problem of neuroscience and practical medicine – diagnostic of central nervous system disorders by the analysis of cerebrospinal fluid.
The interesting information was provided, concerning modern approaches to sensors use in cerebrospinal fluid evaluation.
I have some notes, listed below, which are, in my opinion, must be taken into account for accept of your review for publishing.
The review should be better structured, in my opinion - the data looks somewhat disjointed, the focus is on clinical situations and disease groups, although the review is on biosensors.
The concepts of biosensors and electrochemical sensors, (lines 231-234), as well as optical, piezoelectric (line 296) and even low-power microwave sensors (line 271) are somewhat mixed.
There are clear definitions and distinctive properties of various biosensors, including electrochemical sensors (potentiometric, coulometric, amperometric, etc.), optical sensors, etc.-these should be specified in your review, and the current state of the art on the use of various types of sensors in cerebrospinal fluid research should be highlighted. The description does not pay due attention to the comparison of the advantages of using different biosensors depending on which category the biosensors belong to: immunosensors, enzymatic sensors, peptide based, gene based biosensors.
Table 3 lists some biosensing platforms, a number of them use new materials (graphene, nanoparticles), but it does not say about the advantages of new technologies when used in sensors and electrochemical electrodes.
There is no information about the use of multisensors for cerebrospinal fluid studies, although the use of multisensors is a modern trend.
There are no data comparing the possibilities of CNS disease biomarkers research in cerebrospinal fluid with the works in which CNS disease biomarkers research in other, more accessible biological fluids (blood, saliva) is carried out - this is also one of the modern trends (some articles were published).
There are a number of questions about the distinction between invasive and non-invasive sensors: if we are talking about analyzing cerebrospinal fluid, we cannot consider non-invasiveness as an advantage of sensors, even if we then analyze in vitro, the invasive procedure of lumbar puncture has been performed. If we are talking about implantable sensors, it is unlikely that their use can be less complicated and dangerous than performing a spinal tap, since it requires more extensive intervention (surgery rather than needle use) and more highly specialized personnel (a neurosurgical operating team rather than a neurologist). Implantable biosensors cannot be considered non-invasive, by definition (lines 240-241).
In addition, the question of whether devices for monitoring the patency status of shunts used in the treatment of hydrocephalus are biosensors is debatable, as traditionally a biosensor refers to an analytical device for determining a chemical substance.
There are also a number of stylistic inaccuracies in the review:
Line 10 - it is not quite correct to call cerebrospinal fluid an important fluid - any biological fluids in the body have important functions, just as the fluid cannot be a tool for diagnosis (line 10), rather it is an object for study.
Line 16 - it is stated that biosensors accelerate the determination of biomarkers in the body, especially in cerebrospinal fluid - but this is not quite true, a lot of work deals with the determination of biomarkers in blood, semen, saliva, etc.
The use of the word Biker in line 77 is not clear
The style is broken here, in lines 77-80, the sentence looks cumbersome, besides, not only the working electrode can be modified, but other electrodes, too. The types of electrode modifications need to be clarified in this case.
The statement that spinal puncture is the main method of studying pathologies of the central nervous system (line 108) and that spinal puncture is very painful (line 13) sound exaggerated, because local anesthesia can be used during spinal puncture, and in the clinic of nervous diseases the leading methods are currently neuroimaging methods (computer and magnetic resonance tomography).
Thus, in order to approve the publication of your review, it is necessary to bring the work into compliance with the above comments.
Comments on the Quality of English LanguageThe above mentioned stylistic remarks need to be corrected
Reviewer 2 Report
Comments and Suggestions for Authors
In the manuscript titled “ Advancements in Cerebrospinal Fluid Biosensors: Bridging the 2 Gap from Early Diagnosis to Detection of Rare Diseases”, the author demonstrated advancements in cerebrospinal fluid biosensors. This study contains some interesting findings and are valuable but there are some problems. Therefore, minor revision has to be done before this manuscript could be accepted for publication in the journal. Some comments:
1. In the 1.2 section, the authors need to provide detailed information on current progress in novel non-invasive biomedical devices.
2. There are few references in the past three years.
3. The content in Section 1.3 is too complicated.
Reviewer 3 Report
Comments and Suggestions for Authors
This mini-review by Ghazal Hatami-Fard et al summarized the recent biosensor progress for detecting Cerebrospinal fluid(CSF). The discussion on the technical aspects of CSF biosensors is detailed and informative. And the focus on point-of-care biosensor applications demonstrates a forward-thinking approach to clinical diagnostics. However, the authors should address the following concerns before publishing on Sensors.
1) The introduction section is overly simplified; the author's description of the existing challenges and current state is too vague. The author should consider providing more background and details, including what specific shortcomings the current methods have and what their advantages are. Some details to include are: a) definitions of some abbreviations, such as ELISA, HPLC, etc.; b) line 36: details about the concentration?
2) The second section, "CSF sensing and detection, current state, and progress," being a major part of the document, the author should consider increasing the length of the discussion. Additionally, it is strongly recommended that the author includes necessary figures.
3) in the conclusions and outlook section, it is also recommended that the author avoid general overviews, as they do not seem to inspire or provide useful information to the readers. It is suggested that the author discuss more details and prospects of cutting-edge technologies. For example, could single-molecule techniques potentially become viable detection methods(ACS nano 17 (17), 16369-16395; Sensors and Actuators B: Chemical 338, 129863; )? What are the prevalent challenges in the current detection systems, and what are the possible solutions?
Round 2
Reviewer 1 Report
Comments and Suggestions for Authors
Dear Dr. Hatami-Fard!
Thank you for submitting a revised version of your and Dr. Salzitsa Anastasova-Ivanova manuscript “Advancements in Cerebrospinal Fluid Biosensors: Bridging the Gap from Early Diagnosis to Detection of Rare Diseases.”
I have carefully read the cover letter and all corrections. The manuscript has been deeply revised, all comments and notes have been taken into account and appropriate corrections have been made. I have no comments on the new text of the manuscript. I believe that your work can be recommended for publication.